# Gene flow and species boundaries of the genus *Salmonella*

Marta Cobo-Simón,[1] Rowan Hart,[1] Howard Ochman[1]

**ABSTRACT**  The genus *Salmonella* comprises two species, *Salmonella bongori* and *Salmonella enterica*, which are infectious to a wide variety of animal hosts. The diversity within *S. enterica* has been further partitioned into 6–10 subspecies based on such features as host range, geography, and most recently, genetic relatedness and phylogenetic affiliation. Although *Salmonella* pathogenicity is attributable to large numbers of acquired virulence factors, the extent of homologous exchange in the species at large is apparently constrained such that the species and subspecies form distinct clusters of strains. To explore the extent of gene flow within and among subspecies, and to ultimately define true biological species, we evaluated patterns of recombination in over 1,000 genomes currently assigned to the genus. Those *Salmonella* subspecies containing sufficient numbers of sequenced genomes to allow meaningful analysis—i.e., subsp. *enterica* and *diarizonae*—were found to be reproductively isolated from one another and from all other subspecies. Based on the configuration of genomic sequence divergence among subspecies, it is expected that each of the other *Salmonella* subspecies will also represent a biological species. Our findings argue against the application of prescribed nucleotide-identity thresholds to delineate bacterial species and contend that the Biological Species Concept should not be disregarded for bacteria, even those, like *Salmonella*, that demonstrate complex patterns of species and subspecies divergence.

**IMPORTANCE**  The Biological Species Concept (BSC), which defines species boundaries based on the capacity for gene exchange, is widely used to classify sexually reproducing eukaryotes but is generally thought to be inapplicable to bacteria due to their completely asexual mode of reproduction. We show that the genus *Salmonella*, whose thousands of described serovars were formerly considered to be strictly clonal, undergoes sufficient levels of homologous recombination to be assigned to species according to the BSC. Aside from the two recognized species, *Salmonella enterica* and *Salmonella bongori*, several (and likely all) of the subspecies within *S. enterica* are reproductively isolated from one another and should each be considered a separate biological species. These findings demonstrate that species barriers in bacteria can form despite high levels of nucleotide identity and that commonly applied thresholds of genomic sequence identity are not reliable indicators of bacterial species status.

**KEYWORDS**  *Salmonella*, enteric bacteria, recombination, genomics, speciation

The taxonomy of *Salmonella* has been unwieldy and controversial, beginning with the practice of classifying each serotype as a separate species, creating a situation in which the genus encompassed thousands of species whose relationships to one another were largely unknown. Later revisions that considered the genetic relatedness among strains first partitioned strains into seven subgenera (I, II, IIIa, IIIb, IV, V, and VI) (1), which were subsequently redefined as subspecies (or subspecific groups) following the recommendation that the genus comprised a single species, *Salmonella enterica* (2, 3). Further assessments by DNA hybridization and multilocus enzyme electrophoresis (MLEE) elevated subspecies V to the status of a separate species, and current taxonomy

Address correspondence to Marta Cobo-Simón, marta.cobo@outlook.com.

The authors declare no conflict of interest.

See the funding table on p. 11.

supports the existence of two species within the genus, *Salmonella bongori* and *S. enterica*, the latter of which contains six subspecies (*enterica*, *indica*, *salamae*, *houtenae*, *diarizonae*, and *arizonae*) (4, 5). Based on its genomic distance to other subspecies and its emergence as the outgroup to all other subgroups in phylogenetic analyses (1, 6–8), it has been suggested that subspecies *arizonae* be considered a separate species (9). Moreover, genomic surveys of large collections of isolates indicate that *S. enterica* may contain up to five additional subspecies (9–11) and, according to the List of Prokaryotic names with Standing in Nomenclature, *Salmonella subterranea* would be considered a third *Salmonella* species with a correct and validly published name under the ICNP (https://lpsn.dsmz.de/species/salmonella-subterranea).

The arbitrary and cumbersome manner in which *Salmonella* has been classified and named (12), along with the taxonomic disparities that accompany each revision, suggests that species boundaries within *Salmonella* are not well-defined. Despite the purported clonal nature of *Salmonella* based on MLEE (13–15), sequence comparisons have revealed levels of recombination in *Salmonella* that might allow strain classification based on the Biological Species Concept (BSC), which delineates species based on gene exchange (16). Some authors have viewed homologous recombination as having a possible role in *Salmonella* speciation, such that genomic divergence and recombinational barriers between subspecies have rendered each a separate biological species (17). And although a threshold of ≥95% average nucleotide identity (ANI) is routinely applied to delineate bacterial species (18–20), some models pose that as little as a 2% sequence difference is sufficient to impede homologous exchange, causing *Salmonella* populations to become genetically isolated (17).

To address questions concerning the taxonomic structure of *Salmonella* and to define the true biological species within this genus, we analyzed the patterns of divergence and recombination in more than 1,000 genomes representing each of the *Salmonella* species and subspecies, including *Salmonella subterranea*. We compare several current methods used to classify bacterial genomes—two that rely on homologous recombination to distinguish microbial species (21, 22) and two based on genome-wide identity thresholds (19, 23)—to determine the degree to which the existing taxonomy and nomenclature reflect the actual relationships among members of this genus.

## RESULTS

### Multiple *S. enterica* subspecies are distinct biological species

To delineate species boundaries based on the BSC, we first examined the extent of homologous recombination within and between the phylogroups and named subspecies of *Salmonella*. Recombination was assessed with the *ConSpeciFix* pipeline, which measures the ratio of homoplastic/recombinant ($h$) to non-homoplastic ($m$) polymorphisms across the entire set of core genes shared by the set of genomes being considered (21).

Of the 11 phylogroups (i.e., *Salmonella* species or subspecies) represented in the sample, only *S. enterica* subsp. *enterica* and *S. enterica* subsp. *diarizonae* comprised sufficiently large numbers of genomes to be used as reference lineages for the *ConSpeciFix*. Each of the reference lineages exhibited high $h/m$ values indicative of prevalent recombination among genomes within each subspecies; however, addition of genomes from any of the other phylogroups to either of the reference lineages caused sharp reductions in $h/m$ values (Fig. 1), indicating that each phylogroup is reproductively isolated from the two reference lineages, which should each be considered separate biological species (Fig. 2).

### Recombination barriers despite low divergence

To compare the results obtained by *ConSpeciFix*, which recognized each of the reference lineages as a distinct biological species, to a genome-based similarity metric, we calculated ANI for all pairwise comparisons of 1,142 *Salmonella* genomes and clustered

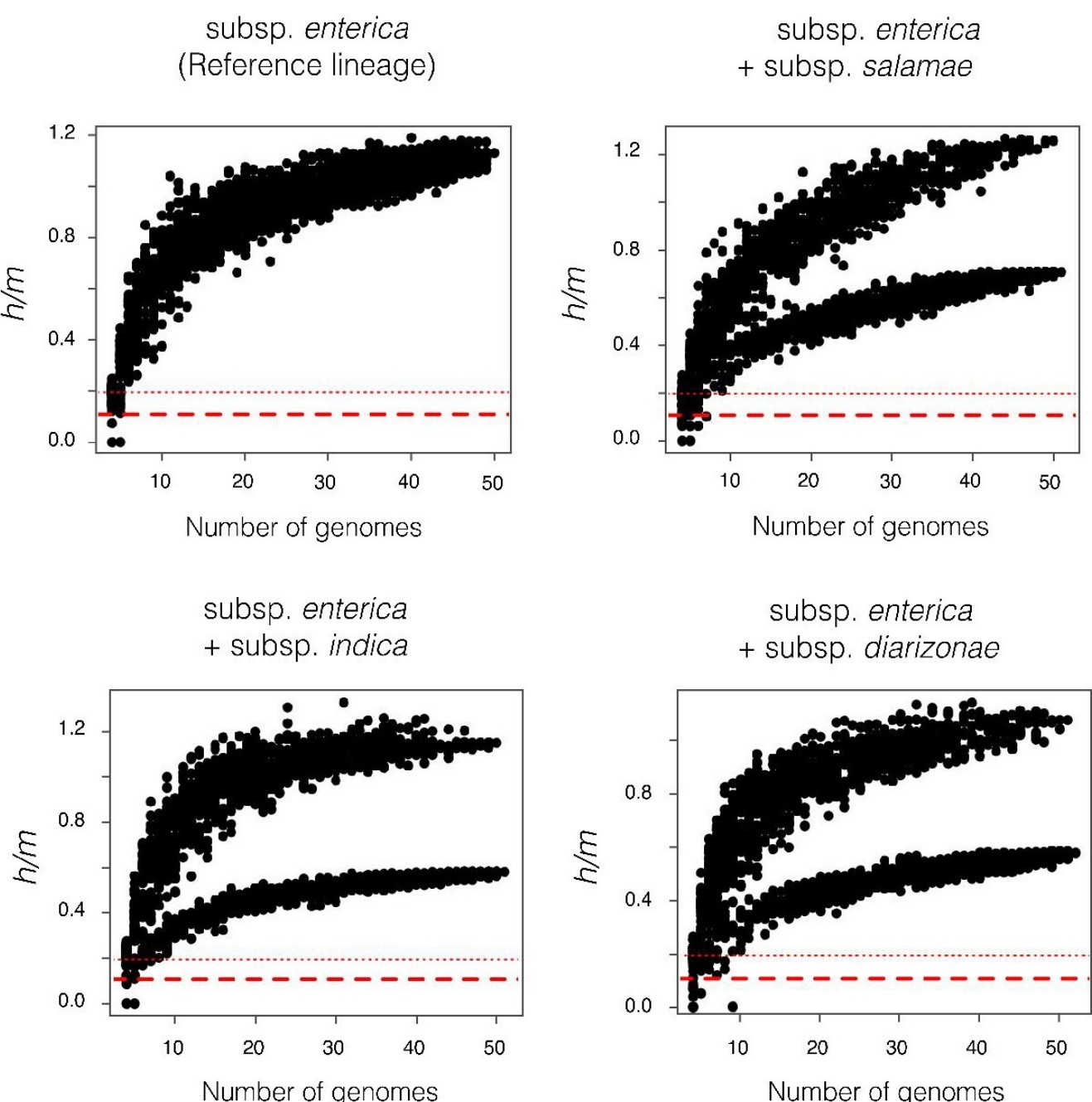

**FIG 1** Measurement of the homoplasies (*h*) and mutations (*m*) ratio of subsamples of *Salmonella* genomes performed by *ConSpeciFix* to distinguish species based on homologous recombination. One curve indicates the presence of a single species, whereas the appearance of a second curve with a lower *h/m* ratio means the absence of recombination and, as consequence, the presence of another biological species. *S. enterica* subsp. *enterica* was used as the reference lineage, which constitutes a single biological species (upper left). Genomes of subspecies *salamae* (upper right), indica (lower left), and *diarizonae* (lower right) were included and each proven to belong to a different biological species. The dashed and dotted red lines in each panel show the average and maximal *h/m* ratios expected when all homoplasies are introduced solely by convergent mutations.

values into groups (ANI-species) at various sequence-identity thresholds (95%–99%). This clustering method separated phylogroups into a distinct ANI-species but at sequence-identity values that do not conform to those that are conventionally applied. For example, applying an ANI ≥ 95%—the most common threshold used to define bacterial species (19)—produced four groups corresponding to *S. subterranea*, *S. bongori*, *S. enterica* subsp. *arizonae*, and the rest of *S. enterica* subspecies. However, raising the

| | enterica | diarizonae | arizonae | houtenae | indica | salamae | novel A | novel B | novel C | VII | Salm. bongori |
|---|---|---|---|---|---|---|---|---|---|---|---|
| enterica | 98.3 ± 0.47 | | | | | | | | | | |
| diarizonae | 95.0 ± 0.08 | 99.4 ± 0.20 | | | | | | | | | |
| arizonae | 93.2 ± 0.10 | 93.8 ± 0.06 | 99.3 ± 0.20 | | | | | | | | |
| houtenae | 94.7 ± 0.08 | 94.8 ± 0.07 | 93.2 ± 0.07 | 99.4 ± 0.09 | | | | | | | |
| indica | 95.5 ± 0.09 | 95.1 ± 0.05 | 93.1 ± 0.06 | 94.7 ± 0.07 | 99.4 ± 0.02 | | | | | | |
| salamae | 95.9 ± 0.13 | 95.8 ± 0.11 | 93.7 ± 0.09 | 95.4 ± 0.10 | 95.9 ± 0.11 | 98.4 ± 0.69 | | | | | |
| novel A | 95.6 ± 0.11 | 94.6 ± 0.07 | 93.1 ± 0.07 | 97.1 ± 0.09 | 94.6 ± 0.05 | 95.2 ± 0.10 | 97.5 ± 0.04 | | | | |
| novel B | 94.5 ± 0.07 | 94.6 ± 0.07 | 92.8 ± 0.06 | 94.0 ± 0.06 | 94.4 ± 0.05 | 95.1 ± 0.08 | 93.9 ± 0.03 | 99.6 ± 0.09 | | | |
| novel C | 95.0 ± 0.08 | 95.4 ± 0.08 | 93.3 ± 0.08 | 94.6 ± 0.06 | 95.2 ± 0.05 | 95.8 ± 0.10 | 94.4 ± 0.03 | 94.6 ± 0.04 | 99.1 ± 0.26 | | |
| VII | 93.6 ± 0.06 | 93.9 ± 0.05 | 92.8 ± 0.04 | 96.8 ± 0.05 | 93.5 ± 0.03 | 94.1 ± 0.07 | 96.6 ± 0.05 | 93.0 ± 0.02 | 93.6 ± 0.04 | 100 | |
| Salm. bongori | 90.2 ± 0.13 | 90.0 ± 0.09 | 89.4 ± 0.07 | 89.8 ± 0.06 | 89.8 ± 0.08 | 90.1 ± 0.10 | 89.9 ± 0.09 | 89.8 ± 0.08 | 90.0 ± 0.10 | 89.5 ± 0.04 | 99.2 ± 0.43 |
| Salm. (Atlantibacter) subterranea | 80.6 | 80.6 | 80.4 | 80.3 | 80.4 | 80.7 | 80.5 | 80.3 | 80.4 | 80.4 | 80.2 |

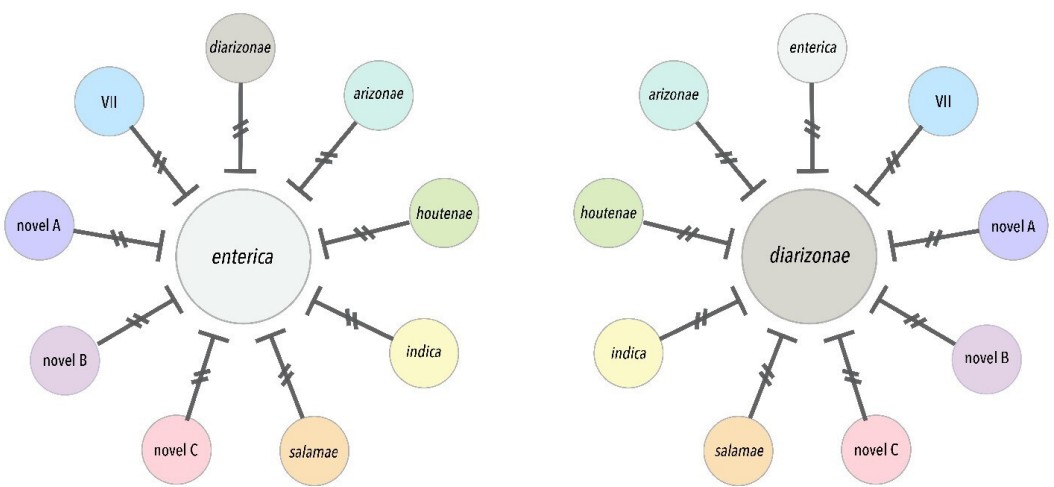

**FIG 2** Nucleotide divergence and biological species status of *Salmonella* subspecies. Matrix of average percent nucleotide identities (ANI values ± SD) within and among the 10 subspecies of *Salmonella enterica*, *Salmonella bongori*, and *Salmonella* (*Atlantibacter*) *subterranea*. Subspecies names are colored as in Fig. 1, and subspecies *enterica* and *diarizonae* are highlighted on account of their use as reference lineages when testing species status using *Conspecifix*. The diagrams (Continued on next page)

**FIG 2** (Continued)

beneath the matrix of ANI values illustrate the reproductive isolation of subspecies *enterica* and *diarizonae*. When either subsp. *enterica* (left) or subsp. *diarizonae* (right) is used as the reference lineage, none of the other nine subspecies of *S. enterica* are found to recombine with either reference lineage, and this interruption of gene flow, indicated by treble crochet (𝄞) symbols, demonstrates that subsp. *enterica* and subsp. *diarizonae* are separate and distinct biological species.

sequence-identity threshold to ≥97% preserved the integrity of *S. subterranea* and *S. bongori* as separate ANI-species but subdivided *S. enterica* into nine discrete, non-overlapping clusters (Fig. 2, left) that comprised from 2 (*S. enterica* subsp. *indica*) to 1,060 genomes (*S. enterica* subsp. *enterica*) (Table S1). *S. subterranea*, which presented an ANI of around 80% in comparison to the other *Salmonella* groups, including *S. bongori*, should be classified as a different genus. These nine groups within *S. enterica* correspond to the named subspecies (and the biological species boundaries defined by *Conspecifix*), except that the subspecies *houtenae* and novel *A* formed a single group. Based on a genome-wide phylogeny of strains, each of these groups is monophyletic (Fig. 3).

There are overt differences between the classification scheme established by *Conspecifix* (and supported by ≥97% ANI clustering) and that presented in the NCBI and GTDB databases (Table S1). The GTDB, which is founded on nucleotide divergence thresholds, distinguishes five species of *Salmonella* (*S. bongori*, *S. enterica*, *S. houtenae*, *S. arizonae*, and *S. diarizonae*) and *S. subterranea* as a different genus (*Atlantibacter subterranea*), whereas NCBI follows classical taxonomy and lists only *S. enterica* and *S. bongori*, as well as *S. subterranea* as a different genus (*A. subterranea*). In the GTDB, *S. bongori* and *S. arizonae* contain only those genomes previously assigned to subspecies *bongori* and *arizonae,* respectively; however, in the GTDB classification scheme: (i) *S. enterica* includes genomes from subspecies *enterica* and *salamae*, (ii) *S. houtenae* includes genomes from subspecies *houtenae*, VII, and novel A, and (iii) *S. diarizonae* includes genomes from subspecies *salamae, diarizonae*, and novel C. The genomes belonging to the subspecies novel B remained unclassified by the GTDB.

## Population dynamics within *S. enterica* subsp. *enterica*

In addition to delineating biological species, the extent of recombination between microbial genomes can reveal their population structures. We applied *PopCOGenT*, which constructs networks of recent gene flow among genomes, to a set of 101 genomes representing *S. bongori* and all of the *S. enterica* subspecies. This set of genomes clustered into 19 "species-groups," 6 of which included the representatives of a single species or subspecies: in particular, *S. bongori* and *S. enterica* subspecies *arizonae*, *diarizonae*, *indica*, *novel B*, and *novel C* each formed their own cluster. One species-group specified by *PopCOGenT* contained representatives of multiple subspecies, encompassing *houtenae*, VII, *novel A*, and two *enterica* genomes. The remaining 12 species-groups were composed of a subset of either *S. enterica* subsp. *enterica* or *S. enterica* subsp. *salamae* genomes, with each of these subspecies divided into six *PopCOGenT* species-groups (Table S1). The species groups obtained with *PopCOGenT* correspond to the species defined by *ConSpeciFix* and ANI except that *PopCOGenT* specifies additional subdivisions within subspecies *enterica* and *salamae*. Differences between the two recombination-based methodologies, and the resolution of supplemental species-groups by *PopCOGenT*, are expected because *PopCOGenT* analyzes entire genomes as opposed to the core set of genes shared by all strains and can define species-groups based on similarities in the accessory genome that arose through lateral gene transfer.

We next investigated the degree of population subdivision within *S. enterica* subsp. *enterica* by applying *PopCOGenT* to genomes representing 20 previously defined ribosomal eBurst groups (reBGs). For this set of 102 subsp. *enterica* genomes, *PopCOGenT* resolved four species-groups containing 76, 12, 8, and 6 genomes, respectively. Although a *PopCOGenT* species-group could include as many as 20 reBGs, there are no cases in which the genomes from a single reBG occupied multiple species-groups (Table S1).

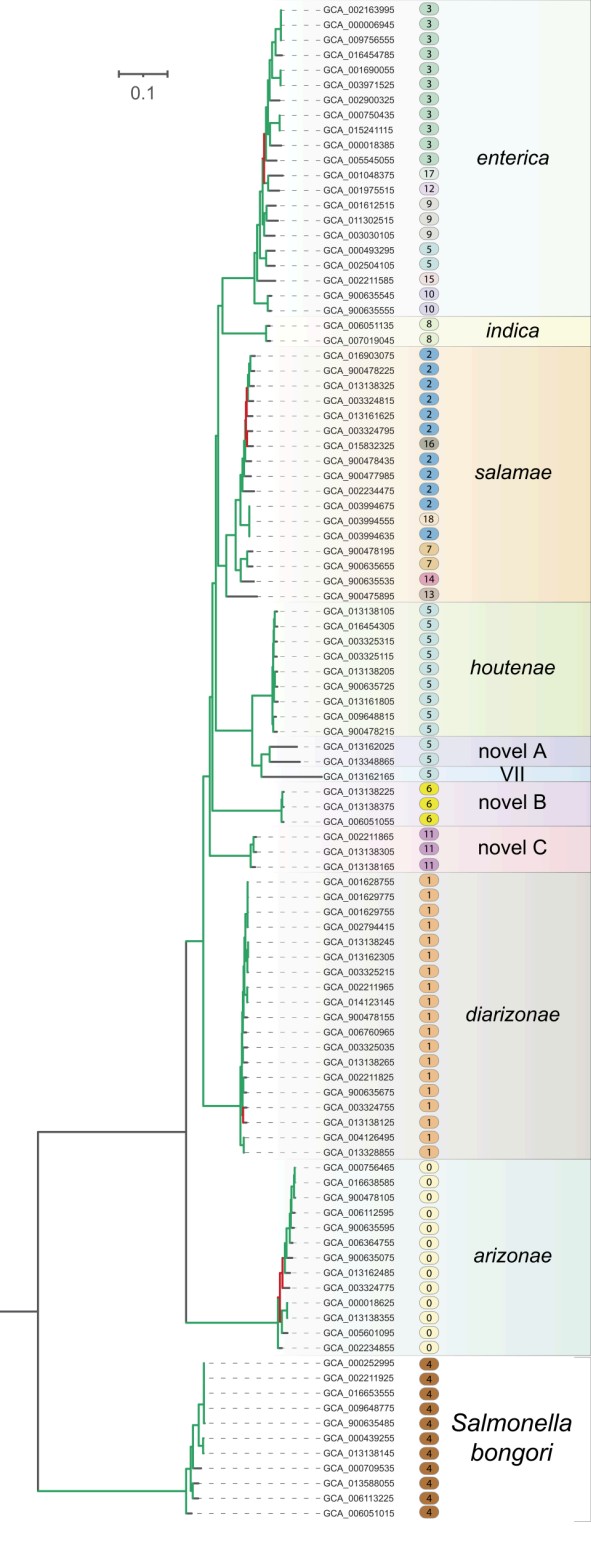

**FIG 3** Phylogenetic relationships of *Salmonella* strains, subspecies, and species. The maximum-likelihood tree is based on 103 genomes selected to represent the diversity in the genus at large. Green branches/nodes have bootstrap support values >90%; red branches/nodes are those with bootstrap support values <60%. Genomes are listed by GCA accession number, followed by a shaded capsule enclosing their *PopCOGenT* species-group number and taxonomic assignment to a *Salmonella enterica* subspecies (or to *Salmonella bongori*). Subspecies names are colored as in Fig. 2.

## DISCUSSION

The enormous strain diversity within *Salmonella* has been partitioned into two species, *S. bongori* and *S. enterica*, the latter of which contains at least 6, or possibly as many as 10, subspecies (4, 5, 8, 24). There have been recommendations that the most divergent subspecies of *S. enterica* (subsp. *arizonae*) be classified a separate species (7, 9), and hints, based on modeling the distribution of genetic variation among strains (17), that perhaps each of the *S. enterica* subspecies represents a distinct species. We have assessed these proposals by using full genome data to establish whether and which *Salmonella* subspecies recombine, thereby resolving species boundaries based on an objective and biologically meaningful criterion.

Among *S. enterica* subspecies, only *enterica* and *diarizonae* currently contain sufficient numbers of sequences for us to accurately assess rates of homologous exchange within and among subspecies; and in each case, the subspecies constitutes a true biological species, distinct from genomes classified to any other subspecies. Each of these biological species is monophyletic, and, based on the branching pattern and levels of nucleotide diversity within and between the named subspecies, it is likely that each of the other *Salmonella* subspecies is a separate biological species. It will require the generation of additional genome sequences to ascertain the species status of some of the subspecies (especially those represented by only one or two genomes), but current sample sizes for *arizonae*, *salamae*, and *houtenae* are approaching those necessary for reliable analysis of site-specific mutation and recombination frequencies. The proposed species *Salmonella subterranean*, formerly assigned to the genus *Atlantibacter*, has been proven to be sexually isolated from the rest of *Salmonella* species and with an ANI low enough to be considered as a separated genus.

The recognition that *Salmonella* subspecies are actually distinct biological species raises questions about the extent of nucleotide sequence divergence necessary to form species boundaries in bacteria. The genomic divergences between *enterica* and *diarizonae*, and the other designated "subspecies" of *Salmonella,* range from 93% to 96% ANI (Fig. 2, left) values that are mostly in line with the current practice of applying a genome-wide ANI threshold of ≥95% to delineate bacterial species (19). The relative ease with which ANI metrics can be used to distinguish groups of bacteria (that often coincide with the established taxonomic nomenclature) has led to its broad appeal and application in many contexts (23, 25, 26). However, the strict adherence to ≥95% ANI to circumscribe bacterial species is ill-advised: for example, strains of *Streptococcus mitis* with an ANI of only 70% constitute a single recombining species (27), which would otherwise be split into >40 "species" based on conventional ANI cutoffs. And in an examination of gene flow across bacterial genomes, over 40% of the BSC-defined bacterial species contain strains that are more than 5% divergent (28).

With regard to the taxonomic status of *S. enterica* subspecies, strict application of a ≥95% ANI threshold would both disrupt current nomenclature and conflict with many of the well-resolved phylogenetic groups on which this nomenclature is grounded. Whereas ANI between subsp. *arizonae* and each of the other subspecies is less than 95% (supporting the recommendation that this subspecies represents a separate species), there are no other cases in which a named subspecies can be consistently merged or separated using this ANI threshold. The narrow range of ANI among *Salmonella* subspecies, and the non-transitivity of values above and below 95% ANI, creates several incongruities. For example, our two focal subspecies, *enterica* and *diarizonae*, are 5% divergent and would therefore be assigned to the same ANI species; however, *enterica* has an ANI of 95.6% to the novel A lineage and *diarizonae* has an ANI of 94.6% to the novel A lineage, rendering species assignment ambiguous. Cases such as these have led to the *post hoc* relaxation of ANI thresholds for some species in order to accommodate the actual similarity among sequenced strains—a practice that both hints to the subjective nature and questions the biological foundations of this metric. It should be noted that in the case of *Salmonella* subspecies, it has been suggested (and confirmed by

the present study) that the species-level ANI threshold be raised to ≥97% to better reflect the patterns of genetic differentiation within and among subspecies (17, 29).

We have argued previously that delineation of bacterial species according to the BSC, such that species membership relies on the capacity for homologous gene exchange, provides a robust classification system that circumvents many of the artificial and arbitrary features inherent to sequence similarity indices (27). The degree of nucleotide identity governs, in part, the efficacy of homologous recombination (30); however, species vary in their capacity for gene exchange at different sequence identity thresholds (31), such that no single value will be applicated across bacterial genomes. Although it was originally thought that asexual lineages were not amenable to classification based on the BSC since clonal individuals are reproductively isolated from one another, upwards of 85% of named bacterial species undergo sufficient levels of homologous exchange to be classified into biological species. On the other hand, methods based on nucleotide identity could be useful to differentiate arbitrary taxa above the species level without any biological meaning.

Our analyses demonstrate that at least two of the *Salmonella* subspecies are valid biological species, but whether *Salmonella* nomenclature is altered, and subspecies designations are eliminated, lies beyond the jurisdiction of the present study. We have recently recommended the addition of a subscripted suffix (e.g., *S. enterica*$_{BIO}$) to indicate members of the same biological species—a practice that would not require immediate changes to existing classification systems (32). Application of a single, universal, and biologically meaningful criterion to define species is appealing because it allows direct comparisons across all lifeforms and will potentially yield new insights into the process of diversification across the tree of Life.

## MATERIALS AND METHODS

### Genomes analyzed

We obtained all complete genome sequences assigned to the genus *Salmonella* in the NCBI RefSeq repository as of March 2021 (https://www.ncbi.nlm.nih.gov/refseq/). We limited our sample to genomes reported in this database to prevent the inclusion of any incomplete or metagenomically assembled genomes, which could obstruct the inference of recombination.

The majority of these 1,142 sequenced genomes were classified as either *S. enterica* ($n = 1121$) or *S. bongori* ($n = 10$); however, 11 genomes were classified as *Salmonella* but not assigned to either of these two species. *S. enterica* is subdivided into 10 subspecies (*enterica, arizonae, diarizonae, salamae, houtenae, indica, VII*, and the novel A, novel B, and novel C subspecies reported by Alikhan et al. [24]). In most cases, the subspecies designation of a strain was specified, but because submission to the NCBI database only requires a species name, subspecific assignments were not reported for 107 of the 1,121 *S. enterica* genomes.

### Data management and strain classification

Because 10% of strains lacked either a species or a subspecies designation, we first clustered and assigned genomes to *S. enterica* subsp. *enterica* based on sequence similarity and then classified the remaining genomes, as follows:

### *Average nucleotide identity*

We computed the ANI, a whole-genome similarity index, between each *Salmonella* genome and the RefSeq reference *S. enterica* subsp. *enterica* sv. Typhimurium strain LT2 using FastANI v1.32 (19). Genomes reporting an ANI ≥ 98% to the reference ($n = 1,055$), a threshold recently applied to distinguish *Salmonella* subspecies (9), were considered members of subspecies *enterica*.

## Maximum-likelihood phylogeny

A set of genomes containing each of the *Salmonella* subspecies ($n = 87$), along with the LT2 reference strain and a random sample of 15 strains from subspecies *enterica* (as identified in the previous step), were subjected to phylogenetic analysis. Genomes were aligned to the LT2 reference genome with NUCmer v3.1 (33) within the PhaME v1.0.2 pipeline (34), and polymorphisms in the set of core genes were used to estimate a maximum-likelihood tree in RAxML v8 applying the GTR + Γ + I model, with branch support appraised by 100 bootstrap replicates (35). The resulting tree consisted of 11 clades, most of which are dominated by strains representing a single *S. enterica* subspecies, and one clade corresponding to *S. bongori* (Fig. 3), thereby allowing the 87 genomes to be taxonomically classified to a species and subspecies according to clade membership. This tree also confirmed three novel clades (A, B, and C) identified in previous analyses (11, 24). (A highly divergent outgroup clade contained two unclassified strains—*Salmonella* sp. HNK130 and *Salmonella* sp. S13—which were subsequently determined to be *Escherichia coli* and removed from further analyses.) The final data set of 1,140 sequenced genomes represented two named species, *S. bongori* ($n = 11$) and *S. enterica* ($n = 1129$), which was subdivided into 10 subspecies: *enterica* ($n = 1060$), *salamae* ($n = 17$), *arizonae* ($n = 13$), *diarizonae* ($n = 19$), *houtenae* ($n = 9$), *indica* (VI) ($n = 2$), VII ($n = 1$), novel A ($n = 2$), novel B ($n = 3$), and novel C ($n = 3$). The complete list of genomes, along with their species and subspecies designations, and NCBI accession numbers, is included in Table S1.

## Distinguishing species boundaries

To determine the integrity of *Salmonella* species and subspecies as currently defined, and to ascertain which genomes should be considered members of the same species, we applied multiple approaches that have been used to define bacterial species based on genome sequences: ANI (19), ConSpeciFix (21), and PopCOGenT (22). ANI applies a prescribed sequence identity threshold (typically ≥95%) to define members of the same species, whereas the latter two methods follow precepts of the BSC and define species as groups whose members exchange genes. Species assignments were compared to those in the Genome Taxonomy Database (GTDB [20]) and EnteroBase (24) (Table S1).

### ConSpeciFix

To ascertain species boundaries as according to the capacity for gene exchange, as proposed by the BSC, we used *ConSpeciFix* v1.3.0, a computational pipeline that searches for recombinant alleles in the core set of genes shared by a set of genomes. For each variable site in the core set of genes, *ConSpeciFix* infers whether the polymorphism is due to recombination, as represented by homoplastic alleles ($h$, i.e., those not inherited vertically from a common ancestor) or to mutation ($m$; those attributable to new or vertically inherited mutations) (21).

Recombination among genomes is measured through a process of subsampling, in which the $h/m$ ratio is calculated for increasing larger sets of subsampled genomes. To test for species membership, a genome sequence (the "test lineage") is added to a set of genomes (the "reference lineages") already known to represent a single biological species. When a test lineage belongs to a different biological species than the reference lineages, the polymorphisms confined to the non-recombining genome(s) lowers $h/m$ ratios, and the plots register either a sharp reduction in $h/m$ ratios or an upper and a lower curve representing subsample iterations with and without the non-recombining genome(s), respectively.

*ConSpeciFix* generally requires the set of reference lineages to contain at least 15 genome sequences; however, due to the low level of diversity in *Salmonella*, this number needed to be increased to $n \geq 19$ sequences to determine species boundaries. Both *S. enterica* subsp. *enterica* and subsp. *diarizonae* satisfied this requirement, and the remaining species and subspecies were used as test lineages. For computational

efficiency and to test the efficacy of our methods, we randomly sampled sets of 50 genomes when using *S. enterica* subsp. *enterica* as the reference lineage using the R-sample function. We serially evaluated the recombinational status of each subspecies, such that each *ConSpeciFix* run consisted of 50 subspecies *enterica*, or all subspecies *diarizonae*, genomes as reference, and one or two genomes of an alternate subspecies as the test lineage.

### PopCOGenT

Like *ConSpeciFix*, *PopCOGenT* delineates species boundaries based on gene exchange but infers recombination by searching for stretches of anomalously high sequence identity between genomes under the assumption that recent transfer events produce an enrichment of identical genomic regions (22). We initially applied *PopCOGenT* to a group of 101 genomes that included: (i) 15 randomly sampled *S. enterica* subsp. *enterica* genomes, (ii) 5 of the most divergent *S. enterica* subsp. *enterica* genomes, (iii) the reference strain *S. enterica* subsp. *enterica* sv. Typhimurium strain LT2, (iv) all complete genomes of *S. bongori* ($n = 11$), and (v) all complete genomes of the remaining 9 *S. enterica* subspecies ($n = 69$). We also applied *PopCOGenT* to a set of 102 subsp. *enterica* sequences representing 20 previously defined ribosomal eBurst groups by EnteroBase (ranging from 2 to 6 sequences per eBurst group) (https://enterobase.warwick.ac.uk/), each of which contains strains whose ribosomal sequence types differ by at least two changes (24).

### Average nucleotide identity

ANI—a whole-genome metric for evaluating the degree of DNA sequence identity between pairs of genomes—was calculated using *FastANI* v1.32 (19). A threshold of ANI ≥ 95% is typically used to partition bacterial sequences into species-level groups (20, 36); however, the level of recombination reported for *Salmonella* (29) advised the use of higher thresholds (e.g., ANI ≥ 97%). To mitigate the non-transitivity of similarity values when genomes are compared to a single reference strain, we performed all pairwise comparisons using the "many-to-many" option of *FastANI* and considered two genomes as members of the same ANI-species if their pairwise ANI was above the prescribed threshold with one another but not with members of another cluster.

Species assignments derived from each of the methods were compared to the taxonomic classifications registered in the GTDB which, such as NCBI, employs an ANI-based approach to classify bacterial genomes to species (i.e., ≥95% ANI to a representative strain). Genomes that were absent from the GTDB website as of September 2021 were analyzed with GTDB-Tk v1.7.0 (23) for GTDB taxonomic classification.

### ACKNOWLEDGMENTS

This work was supported by the NSF Dimensions (grant number 1831730) and NIH (award number R35GM118038) from the National Institutes of Health to H.O.

The authors would like to thank Kim Hammond for assistance in producing the figures.

H.O. conceived the study; H.O. and M.C.-S. supervised the research planning, activity, and execution; M.C.-S. and R.H. performed the computational analyses and analyzed and compiled data; H.O. wrote the paper; M.C.-S. and R.H. read, edited, and approved the final version.

### AUTHOR AFFILIATION

[1]Department of Molecular Biosciences, University of Texas at Austin, Austin, Texas, USA

## PRESENT ADDRESS

Marta Cobo-Simón, Centro de Biotecnología y Genómica de Plantas, Universidad Politécnica de Madrid (UPM) - Instituto Nacional de Investigación y Tecnología Agraria y Alimentaria (INIA-CSIC), Madrid, Spain

Rowan Hart, Department of Ecology and Evolution, University of Chicago, Chicago, Illinois, USA

## AUTHOR ORCIDs

Marta Cobo-Simón ⓘ http://orcid.org/0000-0001-8341-6025

## FUNDING

| Funder | Grant(s) | Author(s) |
| --- | --- | --- |
| National Science Foundation (NSF) | 1831730 | Howard Ochman |
| HHS \| National Institutes of Health (NIH) | R35GM118038 | Howard Ochman |

## AUTHOR CONTRIBUTIONS

Marta Cobo-Simón, Formal analysis, Supervision, Visualization, Writing – review and editing | Rowan Hart, Formal analysis, Visualization, Writing – review and editing | Howard Ochman, Supervision, Conceptualization, Writing – original draft

## DATA AVAILABILITY

All complete genomes used in this analysis are available from the NCBI database using the accession numbers listed in Table S1.

## ADDITIONAL FILES

The following material is available online.

### Supplemental Material

**Table S1 (mSystems00292-23 S0001.xlsx).** Genomes analyzed and their designations by various classification schemes.

### Open Peer Review

**PEER REVIEW HISTORY (review-history.pdf).** An accounting of the reviewer comments and feedback.

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
