## [Reviewer comments · mSystems]

Gene Flow and Species Boundaries of the genus *Salmonella*

Marta Cobo Simón, Rowan Hart, and Howard Ochman

Corresponding Author(s): Marta Cobo Simón, The University of Texas at Austin

Review Timeline:

Submission Date:	March 28, 2023
Editorial Decision:	April 26, 2023
Revision Received:	June 7, 2023
Accepted:	June 7, 2023

Editor: Jack Gilbert

Reviewer(s): The reviewers have opted to remain anonymous.

Transaction Report:

DOI: <https://doi.org/10.1128/msystems.00292-23>

April 26, 2023

Dr. Marta Cobo Simón
The University of Texas at Austin
Molecular Biosciences
2506 Speedway
Austin, Texas 78712

Re: mSystems00292-23 (Gene Flow and Species Boundaries of the genus *Salmonella*)

Dear Dr. Marta Cobo Simón:

Thank you for submitting your manuscript to mSystems. We have completed our review and I am pleased to inform you that, in principle, we expect to accept it for publication in mSystems. However, acceptance will not be final until you have adequately addressed the reviewer comments.

To be honest, we prevaricated between a major or minor revision. ASM does allow both, but one comes with a reject with resubmission (to give you the chance to submit elsewhere without having to withdraw) or minor revision. We were torn between the two. In the end we have gone with the former - but I want to strongly recommend that you consider the revisions requested by both reviewers, or provide rigorous rebuttal. Both reviewers pointed out that the work was very interesting and warranted publication. However, the major concern is that there is a concern that methodological bias may result in the analysis not actually supporting the presented conclusions. There is a concern based on prior publications that the results may not be reproducible, which is a serious concern that needs to be addressed. You should be careful to present all aspects of your pipeline to allow the results to be replicated with the available data. Importantly as well toning down the rhetoric as to the breadth of application of the tool would be appropriate.

Preparing Revision Guidelines

Please return the manuscript within 60 days; if you cannot complete the modification within this time period, please contact me. If you do not wish to modify the manuscript and prefer to submit it to another journal, please notify me of your decision immediately so that the manuscript may be formally withdrawn from consideration by mSystems.

Sincerely,

Jack Gilbert

Editor, mSystems

Journals Department
Reviewer comments:

Reviewer #1 (Comments for the Author):

Review:

The paper "Gene Flow and Species Boundaries of the genus *Salmonella*" suggests that *Salmonella enterica* may be reasonably divided into several species that can be also defined by fulfilling the Biological Species Concept. Moreover, the paper highlights how commonly applied Average Nucleotide Identity (ANI) thresholds may be unreliable in delineating some species. The results reveal population structures within *S. enterica* subsp. *enterica*, and the findings are a robust and well-argued challenge to the classification schemes established by the NCBI and GTDB databases. Overall, the paper contributes significantly to understanding bacterial classification and population structure, providing new insights into the classification of *S. enterica* subspecies (or species?).

Having said that, I do have some concerns about the framing in the last line of the abstract re applying the BSC to "all lifeforms". There can be value in using this approach to salmonella, and possibly to many other bacteria. However this is not the same as arguing that it will necessarily be applicable to all of life. I strongly suggest toning this down. Maybe to something like stating that the BSC should not be discarded for organisms even with as complicated a taxonomy as *Salmonella*, without being seriously considered.

I also think that the MS would be considerably enhanced by some deeper consideration of sampling. This is both important for detecting recombination (hard between closely related taxa if donor and recipient are not both present) and the characteristics of diversity that has yet to be included. Although on this last point, I think it could also be noted that the findings of this work are to a large degree made possible by and reflect the value of increased sampling!

Comments:

#1 - The authors do an excellent job comparing their results with NCBI and GTDB databases. I wonder if the authors were interested in exploring the genus boundaries of *Salmonella*, especially because 95% ANI may not be adequate to delineate species here. According to the List of Prokaryotic names with Standing in Nomenclature (LPSN), *Salmonella subterranea* would be considered a third *Salmonella* species with a correct and validly published name under the ICNP (<https://lpsn.dsmz.de/species/salmonella-subterranea>). Can this be reconciled with the results presented in the paper?

#2 - There is a small error in Table S1 regarding accession numbers highlighted in green. Genbank/RefSeq accession numbers have the prefix "GCA_/GCF", followed by nine numbers, and the suffix ".X", where X usually ranges from 1 to 3. For all accessions highlighted in green in Table S1, the nine-number code remains the same, and the suffix X ranges from 1 to 100, likely an Excel error. This error creates unreal genome accessions and must be corrected to ensure that users can download the data if necessary.

#3 - Regarding the figures in the manuscript, it may be a good idea to bring Figure S1 to the main to demonstrate how *S. enterica* subspecies behave differently regarding h/m. Moreover, in Figure 2, the B panel is unnecessary because both figures capture the same information.

Reviewer #2 (Comments for the Author):

This work explores gene flow within the genus *Salmonella*, with the goal of defining biological species boundaries based on barriers to gene flow, with a particular emphasis on *S. enterica*, which currently comprises several subspecies. I agree with the authors that the current operational definition of microbial species based solely on average nucleotide identity (ANI) is not the best way to define a species and seems arbitrary in a biological sense. Not every species evolves at the same rate, not even having the same ecological limitations.

That being said, I fear that the methodological approach taken by the authors may lead to a systematic bias, in which the recombination-based species definitions are dependent on the set of genomes under consideration.

If we assume that in a natural environment with no ecological or physical boundaries phylogenetically close strains naturally recombine at a higher rate than distantly related strains, the resulting conclusion is that we will always see a (moderate to strong) drop in the h/m ratio between distant groups. That will mean that a pre-clustering of the strains into well defined, phylogenetic coherent groups will naturally result in drops of h/m ratios between the groups, especially in distant ones. Of course, the co-occurrence of strains in the same environment or the absence of ecological barriers cannot be directly assumed. For example, ecological isolation could be expected in a pathogenic species such as *Salmonella*.

My main concern then is that the approach is open to methodological bias depending on the genomes selected for analysis, in this case suggesting that multiple biological species exist within the *S. enterica* group. The PopCoGenT analysis seems to point in this direction too by further subdividing the enterica subspecies. By contrast, the study of Bobay and Ochman in 2017 using the same methodology in which 64 *S. enterica* genomes were used (including several subspecies that have been clearly separated into new species in this paper - please see supplementary material in 2017 paper) found a single species curve, i.e. no interruptions in the gene-flow were identified. This strongly suggests that the definition of biological species using this approach may not have a binary "yes-or-no" answer. The results may be more akin to a fractal, in which different groups are defined as species depending on the genomes used. That may indeed have biological sense, as species naturally diverge and may be on the way to speciation... ¿but can they be considered as new species? This could perhaps be directly tested by random resampling of the *Salmonella* dataset to see how robust the biological species are (instead of guiding the analysis to predefined groups).

Other issues may be found in line-to-line revision:

102-104: A search in NCBI for *Salmonella* assemblies released before March 2021 revealed more than 10000 assemblies. With what criteria were these 1142 genomes sampled? Was it a quality-based election, taxonomy metadata existence based?

Please, specify.

103-104: "Various sequence-identity threshold" sounds generic.

104-106- Was the Alignment fraction included as a variable in the ANI analysis?

104-106-The ANI methods clearly explain how the subspecies enterica was defined, but not how the rest of the species clusters were defined. Were they compared against the subspecies types?

111: Please review Supplementary table 1, all the 100 first genomes have the same accession code

114- phylogeny. Why subsample 87 strains for the phylogenetic analysis? What were the criteria for the selection? Why not make a complete phylogeny of the whole database?

141- So it appears that PopCOGenT deliver different groups changing the database, which is interesting. Has ConSpeciFix been tested in the same way?

171- I think (21) is not the correct reference, as no *Streptococcus mitis* data can be found in this paper. Maybe the paper referenced is Bobay and Ochman 2017? (As some of the authors are also in this paper, you are free to correct me, but I could not have access to the data though this paper). Anyhow, the genomes studied in 2017 were always above 80% ANI. It's still quite extreme value, but not 70%.

174- Is (12) reference relevant for this statement?

174-176- "And recently, a comprehensive examination of gene flow across bacterial genomes showed that over 80% of the >2600 species analyzed contained strains whose ANI values are lower than 95% (27)." - I think this paper does not sustain this statement...

275-280- Please, see general comments for the review on the methodological approach and possible bias. Also, I am very concerned about the reproducibility of this method. "For computational efficiency, we randomly sampled sets of 50 genomes when using *Salmonella enterica* subsp. *enterica* as the reference lineage." How was that made? And how the output was integrated to obtain the final results? The phrasing is generic. The method to analyze a large amount of data is of crucial interest, however it can't be directly inferred from the text and the ConSpeciFix tool v1.3.0 available on GitHub. Maybe the repository could be updated to reflect the updates in the pipeline.

How the n{greater than or equal to}19 reference sequences are chosen? Are they chosen randomly in the enterica ANI defined species for each subsample? Will the size of the subset database matter for the analysis?

As ConSpeciFix relies on the calculation of a core genome alignment, if the ConSpeciFix tool is relaunched de-novo each time it is run, won't that affect the initial core alignment and the definition of the dominant variant for the h/m calculation?

Even if it is not "computationally efficient", is it possible to analyze all the strains in the database in the same analysis? Will the result show several curves like the ones shown in Figure 1 (but in a single graph)?

281: It would be very interesting to see if the results with (only) this subset of genomes match in ConSpeciFix and PopCOGenT. As for the results, it appears that the results of PopCOGenT rather subdivide the subspecies defined by ConSpeciFix in several

groups, but this fact is not really discussed. Is there a reason?

424- Why use average identities in figure 2? Won't be more useful to give the minimum in each for the definition of the species?
Or at least reflect the standard deviation of this value.

We thank the Editor for handling our manuscript. In this new version, we address all points raised by the Reviewers, perform several new analyses, amend our figures, and revise various sections of the text. Below, we list the reviewers' comments (*italics*) and our responses (Roman), and explain how the manuscript has been changed to accommodate their concerns.

Reviewer 1.

The paper "Gene Flow and Species Boundaries of the genus Salmonella" suggests that Salmonella enterica may be reasonably divided into several species that can be also defined by fulfilling the Biological Species Concept. Moreover, the paper highlights how commonly applied Average Nucleotide Identity (ANI) thresholds may be unreliable in delineating some species. The results reveal population structures within S. enterica subsp. enterica, and the findings are a robust and well-argued challenge to the classification schemes established by the NCBI and GTDB databases. Overall, the paper contributes significantly to understanding bacterial classification and population structure, providing new insights into the classification of S. enterica subspecies (or species?).

Having said that, I do have some concerns about the framing in the last line of the abstract re applying the BSC to "all lifeforms". There can be value in using this approach to salmonella, and possibly to many other bacteria. However this is not the same as arguing that it will necessarily be applicable to all of life. I strongly suggest toning this down. Maybe to something like stating that the BSC should not be discarded for organisms even with as complicated a taxonomy as Salmonella, without being seriously considered.

Thank you for the suggestion. We admit that in the context of the current manuscript, this was an overstatement, so we have revised the Abstract accordingly (lines 37–39).

I also think that the MS would be considerably enhanced by some deeper consideration of sampling. This is both important for detecting recombination (hard between closely related taxa if donor and recipient are not both present) and the characteristics of diversity that has yet to be included. Although on this last point, I think it could also be noted that the findings of this work are to a large degree made possible by and reflect the value of increased sampling!

Whereas we agree that while more is usually better, in this case, increased sampling will not affect the results. The set of genomes that we analyzed exposed recombination among members of each subspecies, and the inclusion of additional strains cannot erase a signal of recombination. With regard to the barriers between subspecies that we identified, unless there are genomes that are somehow anomalous and misclassified, the barriers will remain.

Comments

1. The authors do an excellent job comparing their results with NCBI and GTDB databases. I wonder if the authors were interested in exploring the genus boundaries of Salmonella, especially because 95% ANI may not be adequate to delineate species here. According to the List of Prokaryotic names with Standing in Nomenclature (LPSN), Salmonella subterranea would be considered a third Salmonella species with a correct and validly published name under the ICNP (<https://lpsn.dsmz.de/species/salmonella-subterranea>). Can this be reconciled with the results presented in the paper?

The revised manuscript includes the *Salmonella (Atlantibacter) subterranea* genome (see Figure 2).

2. There is a small error in Table S1 regarding accession numbers highlighted in green. Genbank/ RefSeq accession numbers have the prefix "GCA_/GCF", followed by nine numbers, and the suffix ".X", where X usually ranges from 1 to 3. For all accessions highlighted in green in Table S1, the nine-number code remains the same, and the suffix X ranges from 1 to 100, likely an Excel error. This error creates unreal genome accessions and must be corrected to ensure that users can download the data if necessary.

Thank you for bringing this to our attention. The table has been corrected.

3. Regarding the figures in the manuscript, it may be a good idea to bring Figure S1 to the main to demonstrate how S. enterica subspecies behave differently regarding h/m. Moreover, in Figure 2, the B panel is unnecessary because both figures capture the same information.

Thank you for the suggestions. We are generally circumspect about which figures should be included in the manuscript and which are relegated to the Supplementary material, but we would be happy to move Figure S1 to the body of the text. It now serves as Figure 1 in the revised version.

The matrix in Figure 2 provides ANI values but does not indicate if there is gene flow between subspecies, so we include (but shrink) the lower panel, which clearly illustrates the disruption of gene flow among subspecies and assists in the interpretation of values in the matrix.

Reviewer 2

This work explores gene flow within the genus Salmonella, with the goal of defining biological species boundaries based on barriers to gene flow, with a particular emphasis on S. enterica, which currently comprises several subspecies. I agree with the authors that the current operational definition of microbial species based solely on average nucleotide identity (ANI) is not the best way to define a species and seems arbitrary in a biological sense. Not every species evolves at the same rate, not even having the same ecological limitations.

That being said, I fear that the methodological approach taken by the authors may lead to a systematic bias, in which the recombination-based species definitions are dependent on the set of genomes under consideration.

If we assume that in a natural environment with no ecological or physical boundaries phylogenetically close strains naturally recombine at a higher rate than distantly related strains, the resulting conclusion is that we will always see a (moderate to strong) drop in the h/m ratio between distant groups. That will mean that a pre-clustering of the strains into well defined, phylogenetic coherent groups will naturally result in drops of h/m ratios between the groups, especially in distant ones. Of course, the co-occurrence of strains in the same environment or the absence of ecological barriers cannot be directly assumed. For example, ecological isolation could be expected in a pathogenic species such as Salmonella.

My main concern then is that the approach is open to methodological bias depending on the genomes selected for analysis, in this case suggesting that multiple biological species exist within the S. enterica group. The PopCoGenT analysis seems to point in this direction too by further subdividing the enterica subspecies. By contrast, the study of Bobay and Ochman in 2017 using the same methodology in which 64 S. enterica genomes were used (including several subspecies that have been clearly separated into new species in this paper - please see supplementary material in 2017 paper) found a single species curve, i.e. no interruptions in the gene-flow were identified. This strongly suggests that the definition of biological species using this approach may not have a binary "yes-or-no" answer. The results may be more akin to a fractal, in which different groups are defined as species depending on the genomes used. That may indeed have biological sense, as species naturally diverge and may be on the way to speciation... ¿but can they be considered as new species? This could perhaps be directly tested by random resampling of the Salmonella dataset to see how robust the biological species are (instead of guiding the analysis to predefined groups).

Thank you very for bringing up this issue. Having checked the supplementary material of the 2017 paper, we see that all but one of the *Salmonella enterica* genomes belonged to the subspecies *enterica*, which is why they were classified as same species. This mirrors our results, and there does not appear to be a sampling bias, at least in this case.

Random resampling certainly seems to be the best approach, which is why we subsampled 50 *S. enterica* subsp. *enterica* for use as reference genomes for these analyses (see Methods, line 304).

Other issues may be found in line-to-line revision:

1. Lines 102-104: A search in NCBI for Salmonella assemblies released before March 2021 revealed more than 10000 assemblies. With what criteria were these 1142 genomes sampled? Was it a quality-based election, taxonomy metadata existence based? Please, specify.

The criteria by which we selected this set of genomes for analysis are explained (lines 239-241).

2. Lines 103-104: "Various sequence-identity threshold" sounds generic.

The range of thresholds is now specified in the text (line 110).

3. Lines 104-106- Was the Alignment fraction included as a variable in the ANI analysis?

We found that this was not necessary since the genomes were highly similar and we limited our analyses to our complete genomes.

4. Lines 104-106-The ANI methods clearly explain how the subspecies *enterica* was defined, but not how the rest of the species clusters were defined. Were they compared against the subspecies types?

We are sorry if this was unclear. As described in the Methods section, we performed a pre-analysis (based on ANI) to distinguish genomes belonging to the subspecies *enterica* and classified the rest of genomes phylogenetically, which partitioned the different subspecies into different clades. Phylogroups were then classified based on ANI (Table S1) and subjected to mutational/recombinational analysis.

5. Lines 111: Please review Supplementary table 1, the 100 first genomes have the same accession code

Indeed they do – the table has been edited and now contains the correct accession numbers.

6. Lines 114- phylogeny. Why subsample 87 strains for the phylogenetic analysis? What were the criteria for the selection? Why not make a complete phylogeny of the whole database?

Due to the high number of genomes analyzed in this study, a phylogeny including all strains would be both difficult to visualize and impractical for algorithms, such as RAxML or IQ-Tree. The tree contains all analyzed genomes for each of the subspecies, except for the subspecies *enterica*, for which we selected a representative set of 21 genomes. Additionally, use of this number of genomes allows direct comparisons between the strain relationships generated by different methods.

7. Line 141- So it appears that PopCOGenT deliver different groups changing the database, which is interesting. Has ConSpeciFix been tested in the same way?

ConSpeciFix does not deliver different groups and is not as sensitive to changes in the database because it is based on the core genome shared by all strains. As explained in the text (lines 148–155), PopCOGenT can be impacted by regions that are confined to a limited number of genomes.

8. Line 171- I think (21) is not the correct reference, as no *Streptococcus mitis* data can be found in this paper. Maybe the paper referenced is Bobay and Ochman 2017? (As some of the authors are also in this paper, you are free to correct me, but I could not have access to the data though this paper). Anyhow, the genomes studied in 2017 were always above 80% ANI. It's still quite extreme value, but not 70%.

Thank you for the suggestion: the reference has been changed.

9. Line 174- Is (12) reference relevant for this statement?

The reference has been removed.

10. Lines 174-176- "And recently, a comprehensive examination of gene flow across bacterial genomes showed that over 80% of the >2600 species analyzed contained strains whose ANI values are lower than 95% (27)." - I think this paper does not sustain this statement...

The statement has been rewritten and its attribution changed to reflect the information presented in figure 2 of Bobay and Ochman (2017).

11. Lines 275-280- Please, see general comments for the review on the methodological approach and possible bias. Also, I am very concerned about the reproducibility of this method. "For computational efficiency, we randomly sampled sets of 50 genomes when using *Salmonella enterica* subsp. *enterica* as the reference lineage." How was that made? And how the output was integrated to obtain the final results? The phrasing is generic. The method to analyze a large amount of data is of crucial interest, however it can't be directly inferred from the text and the ConSpeciFix tool v1.3.0 available on GitHub. Maybe the repository could be updated to reflect the updates.

The subsampling of genomes for the reference lineages were randomly generated, using overlapping subsamples from the entire set of genomes. For test lineages, we used the two most divergent strains from each of the subspecies. Because all of the results were congruent and produced the same gene flow barriers, we conclude that each of the subspecies of *S. enterica* constitutes a different biological species.

12. How the was $n \geq 19$ reference sequences are chosen? Are they chosen randomly in the enterica ANI defined species for each subsample? Will the size of the subset database matter for the analysis?

In our experience using *Conspecifix* (on several hundred species), a minimum of 15 strains yields accurate results, and in the present study, only subspecies *enterica* and *diarizonae* met this requirement. For *diarizonae*, we use all available genomes ($n = 19$), and for *enterica*, we randomly selected subsets of 50 strains.

13. As *ConSpeciFix* relies on the calculation of a core genome alignment, if the *ConSpeciFix* tool is relaunched de-novo each time it is run, won't that affect the initial core alignment and the definition of the dominant variant for the h/m calculation?

Tests using randomly sub-sampled sets of strains all yield the same results, so the method is robust to core genome alignments built from different sets of strains.

14. Even if it is not "computationally efficient", is it possible to analyze all the strains in the database in the same analysis? Will the result show several curves like the ones shown in Figure 1 (but in a single graph)?

It is possible to include all strains of the same reference lineage in a single analysis if they all of belong to the same species. However, all the strains in the database constitute multiple "reference lineages" and contain a mix of very divergent genomes, so the allelic diversity will potentially be high and yield a large spread of points at each sampling iteration (thereby, masking the detection of different species). For this reason, we first tested if all *S. enterica* subsp. *enterica* genomes (or all *diarizonae* genomes) belonged to the same species and then included various test lineages.

15. Line 281: It would be very interesting to see if the results with (only) this subset of genomes match in *ConSpeciFix* and *PopCOGenT*. As for the results, it appears that the results of *PopCOGenT* rather subdivide the subspecies defined by *ConSpeciFix* in several groups, but this fact is not really discussed. Is there a reason?

The 101 strains analyzed with *PopCOGenT* are shown in the Figure 3, which compares the results of ANI and *ConSpeciFix* on the same strains. The results of the analyses match, except that in both subspecies *enterica* and *salamae*, *PopCOGenT* distinguishes some additional sub-species that remain discrete from other subspecies. The reasons why these methods might yield difference results are now included in the script (lines 148–155).

16. Line 424- Why use average identities in figure 2? Won't be more useful to give the minimum in each for the definition of the species? Or at least reflect the standard deviation of this value.

Thank you for the suggestion. We include standard deviations underneath the ANI values in Figure 2.

June 7, 2023

Dr. Marta Cobo Simón
The University of Texas at Austin
Molecular Biosciences
2506 Speedway
Austin, Texas 78712

Re: mSystems00292-23R1 (Gene Flow and Species Boundaries of the genus *Salmonella*)

Dear Dr. Marta Cobo Simón:

Your manuscript has been accepted, and I am forwarding it to the ASM Journals Department for publication. For your reference, ASM Journals' address is given below. Before it can be scheduled for publication, your manuscript will be checked by the mSystems production staff to make sure that all elements meet the technical requirements for publication. They will contact you if anything needs to be revised before copyediting and production can begin. Otherwise, you will be notified when your proofs are ready to be viewed.

If you would like to submit a potential Featured Image, please email a file and a short legend to mSystems@asmusa.org. Please note that we can only consider images that (i) the authors created or own and (ii) have not been previously published. By submitting, you agree that the image can be used under the same terms as the published article. File requirements: square dimensions (4" x 4"), 300 dpi resolution, RGB colorspace, TIF file format.

We recognize that the video files can become quite large, and so to avoid quality loss ASM suggests sending the video file via <https://www.wetransfer.com/>. When you have a final version of the video and the still ready to share, please send it to mSystems staff at mSystems@asmusa.org.

Sincerely,

Jack Gilbert
Editor, mSystems

Journals Department
E-mail: mSystems@asmusa.org